Effects of defeat and entrapment on suicide risk in university population: the role of rumination and sex

Silvestre Vidal Inmaculada Nayara 1
Nieto Marta 1 marta.nieto@uclm.es
Ricarte Jorge 1
Vizcaíno Alcantud María Dolores 1
Hallford David 2
http://orcid.org/0000-0001-5567-8589 Ros Laura 1
1 Departament of Psychology/Faculty of Medicine, University of Castilla-La Mancha , Albacete , Spain
2 Department of School of Psychology/Faculty of Health, Deakin University , Melbourne , Australia
Epifania Ottavia M.
Electronic publication date: 2024 Dec 11
Publication date: 2024
Volume: 12
Electronic Location ID: e18673
Received 2024 May 14; Accepted 2024 Nov 19
Copyright: © 2024 Silvestre Vidal et al.
Copyright year: 2024
Copyright holder: Silvestre Vidal et al.
License: This is an open access article distributed under the terms of the Creative Commons Attribution License, which permits unrestricted use, distribution, reproduction and adaptation in any medium and for any purpose provided that it is properly attributed. For attribution, the original author(s), title, publication source (PeerJ) and either DOI or URL of the article must be cited.
License URL: https://creativecommons.org/licenses/by/4.0/

Keywords: Suicide risk, Defeat, Entrapment, Rumination, University students

Funding: Castilla-La Mancha Department of Education Culture and Sports and the European Regional Development Fund SPBLY/19/180501/000181 This study has been supported by the Castilla-La Mancha Department of Education, Culture and Sports and the European Regional Development Fund under SPBLY/19/180501/000181 grant. The funders had no role in study design, data collection and analysis, decision to publish, or preparation of the manuscript.

==============================
Background

Suicide is recognized as a significant public health issue, with adolescents/young people being a risk group of concern. Taking the integrated motivational-volitional model as a reference, this study focuses on analyzing the association between defeat and entrapment, on the one hand, and suicide risk, on the other, in a university population without depressive symptoms, while also considering the role of rumination and sex.

Method

The survey study involved a community sample of 650 Spanish university students. A total of 524 students (150 men (Mage = 20.6, SD = 3.7) and 374 women (Mage = 20.4, SD = 3.6)) completed self-report measures of suicide risk, defeat, entrapment, and rumination.

Results

Defeat and entrapment were significantly associated with suicide risk, and higher rumination was both directly and indirectly associated with higher levels of entrapment through the variable of defeat. However, the effect of rumination on entrapment varied by sex.

Conclusion

Despite being a cross-sectional preliminary study, this work identifies important variables in the trajectory of suicidal ideation. Adolescence and young adulthood are a critical stage for intervening to reduce the risk of death by suicide, and this study provides findings that may inform preventive approaches.

Introduction

The World Health Organization recognizes suicide as a high priority for public health policies (World Health Organization, 2021), with university students being a risk group of particular concern (Han et al., 2018). According to the Spanish National Institute of Statistics (2024), the suicide rate in Spain is around five per 100,000 inhabitants for young people aged 15 to 29 years, with marked differences in terms of sex: more males die by suicide than females (of the total number of people who committed suicide in 2023, 74.7% were men and 25.3% were women). In addition, a study conducted in 16 universities in Spain revealed that approximately one in three participants reported suicidal ideation over their lifetime (Reina-Aguilar, Díaz-Jiménez & Caravaca-Sánchez, 2023). In this case, there are also differences by sex: of the university students reporting suicidal ideation, 76.7% were women, while 22.6% were men. These data support the sex paradox in suicidology, which points to a preponderance of females in non-fatal suicidal ideation/behavior and a preponderance of males in completed suicide (Miranda-Mendizabal et al., 2019).

Suicidal thoughts and behaviors among young people, relative to other age groups, are of particular concern for several reasons (see Cha et al., 2018, for a review). First, suicide is one of the leading causes of death during youth compared to other life stages. Second, many individuals that have considered or attempted suicide did so for the first time before the age of 20. Third, suicidal ideation is exacerbated by the increase in academic stress during adolescence (Okechukwu et al., 2022). Fourth, according to the World Health Organization (2021), worldwide, the adolescent and young adulthood population (15–29 age group) has experienced a significant increase in suicide deaths since the mid-2000s, being particularly evident since 2010. This increase varies, however, depending on the region/country. Specifically, in Spain, since 2018, there has been an upward trend in suicide mortality rates, with a 32.35% increase in the number of suicides in this age group between 2019 and 2021. Finally, this is a life period when patterns of maladaptive thoughts, feelings and behaviors may emerge and it is therefore a critical stage for intervening in this phenomenon, and from a preventive approach (McKay et al., 2023; Milbourn et al., 2022; Roza et al., 2023). For these reasons, it is imperative to evaluate the applicability of an integrated model to provide a theoretical basis for understanding the psychological processes underlying suicide and to determine the factors related to suicide in this age demographic.

In regard to integrated models, the integrated motivational-volitional model has provided compelling theoretical and empirical advances in recent times (IMV model; O’Connor, 2011; O’Connor & Kirtley, 2018). The IMV model is a tri-partite model that describes the appearance of suicidal ideation and its progression towards suicide attempts. The first phase is the so-called pre-motivational, which focuses on factors that serve as a reference to determine the suicidal risk. The second phase, referred to as motivational, models suicidal ideation, establishing a relationship between feelings of defeat (i.e., perception of failed struggle, humiliation, or loss of social status) and entrapment (i.e., desire to escape from an unbearable situation, linked to the perception that all routes of escape are blocked). Finally, the volitional phase posits a series of factors that complete the transition from ideation to suicide attempts (e.g., access to lethal means).

Further nuances can be considered with the concept of entrapment, which is a central process of the IMV model, and has transdiagnostic relevance (Siddaway et al., 2015). In particular, the distinction between internal entrapment, which refers to the feeling of being trapped by one’s own thoughts and feelings, and external entrapment, which describes the feeling of being trapped by external circumstances (Owen et al., 2018). Although the IMV model does not distinguish between internal and external entrapment (O’Connor & Kirtley, 2018), a growing number of studies indicate there is value in considering these two separate dimensions of entrapment (e.g., Höller et al., 2022; Moscardini et al., 2022).

The IMV model also identifies a set of motivational moderators that may increase or decrease the likelihood of suicide emerging (O’Connor, 2011; O’Connor & Kirtley, 2018). Rumination refers to a tendency to repetitively and passively focus on one’s distress and the causes, meanings, and consequences of this distress (Nolen-Hoeksema, 1991). According to the IMV model, rumination increases the sense of entrapment leading to the consideration of suicide as a possible escape route, facilitating the transition from suicidal ideation to suicide attempts (Morrison & O’Connor, 2008; Rogers et al., 2017). Different types of rumination, such as depressive rumination, anger rumination, and suicide rumination, can contribute to this process. For this reason, several studies suggest that rumination may play a key role in the defeat-entrapment relationship, and the content of such rumination may be varied (Adler et al., 2015; Griffiths, 2014; Tucker, O’Connor & Wingate, 2016). By extension, it has been suggested that reducing rumination could contribute to an individual experiencing less internal entrapment and help to mobilize their efforts to address problems related to a sense of external entrapment (Sandford et al., 2022).

Sex may be another moderator in these relationships. The so-called sex paradox (Shelef, 2021) has evidenced that females have an almost two-fold higher risk of suicide attempts than males, while males have an almost threefold higher risk of dying by suicide than females (Miranda-Mendizabal et al., 2019). However, these differences remain inconclusive for some of the components of the IMV model. For example, while some studies have found that females report higher levels of entrapment than males (Chu et al., 2017; Cramer, Rasmussen & Tucker, 2019; Ren et al., 2019), others have reported mixed results or no sex-related differences for defeat and entrapment (Cramer, Rasmussen & Tucker, 2019; Lucht et al., 2020). For example, recent research has suggested there are sex differences in the entrapment variable: men present greater generalized entrapment in various aspects of life, while women tend to experience more focused entrapment in expressions of intense emotional change (Ramos-Vera et al., 2024). Meanwhile, given the findings on sex and rumination, namely that from adolescence on, females show a tendency to ruminate more than men (e.g., Espinosa, Martin-Romero & Sanchez-Lopez, 2022; Johnson & Whisman, 2013), there may be interacting effects of these variables on entrapment.

This study

Although the relationship between defeat, entrapment and rumination proposed by the IMV model has been well established (for a review see Souza et al., 2024), there are fewer studies analyzing the relationship between these variables and suicidal ideation in community populations without clinical depressive pathology or without depressive symptomatology (e.g., Dhingra, Boduszek & O’Connor, 2016). In this line, for example, the study by Teismann & Forkmann (2017), conducted with two samples of German population (one community group and one outpatient one), found that perception of entrapment fully mediated the association between rumination and suicide ideation. The authors did not analyze the defeat variable, however. Additionally, the study by Rogers & Joiner (2018), carried out on a community sample in the United States who reported a lifetime history of suicidal ideation, found that suicide-specific rumination was both directly and indirectly associated, through defeat, with the presence of suicide attempts, although the role of entrapment in this relationship was not assessed in the study. Meanwhile, the study by Li et al. (2021), in a sample of Chinese adolescents, reported that, in contrast to the ideas proposed in the IMV model, rumination did not moderate the relationship between defeat and entrapment. As underlined by Dhingra, Boduszek & O’Connor (2016), there is a need to test the IMV theory and replicate the findings in other populations. For this reason, the present study focused on a non-clinical university student sample with the main aim of identifying risk factors and underlying mechanisms of suicide risk in order to gain a broader understanding of suicide risk. Identifying risk factors before clinical disorders emerge is crucial for developing early preventive interventions, and prevention is especially important in the case of suicide, given that many individuals who die by suicide have not been diagnosed or received professional care (Tang et al., 2022).

Therefore, the present study seeks to contribute to the suicide prevention literature by identifying risk factors in a non-clinical population. To this end, the relationship proposed by the IMV model between the variables of rumination, defeat, entrapment, and suicidal risk is assessed in a sample of Spanish university students without depressive symptoms. We propose the following aims: (1) to analyze the possible direct and indirect associations between defeat and suicide risk through the entrapment variable. In this sense, as suggested by the previous scientific literature, it is expected to find that defeat shows both a direct and an indirect effect on suicide risk through both types of entrapment; (2) to assess the degree to which rumination acts as a moderator of the relationship between the variables of defeat and entrapment (both external and internal). Drawing on previous findings, it is expected to find a moderating effect of rumination on the relationship between defeat and both types of entrapment, such that rumination strengthens the positive relationship between defeat and entrapment; and (3) in an explorative manner, to perform a multi-group comparison to determine whether the relationships identified between these variables differ according to sex.

Materials and Methods

Participants

A convenience sample of participants was used, comprising 650 Spanish university students, enrolled across all the branches of learning studied at public universities. The inclusion criteria were: (1) absence of diagnosed psychological disorders (self-reported); (2) absence of clinically significant depressive symptoms. This variable was assessed using the Patient-Reported Outcomes Measurement Information System-Depression (PROMIS®; Cella et al., 2010), with a score >16 being taken as an indicator of the presence of clinically significant depressive symptoms; and (3) giving signed informed consent. Using these criteria, 126 participants were excluded from the study (see Fig. 1). The final sample included a total of 524 students, 150 men (ages 17–51 years; M = 20.6, SD = 3.7) and 374 women (ages 17–52 years; M = 20.4, SD = 3.6). None of these participants reported previous suicidal behavior.

Figure 1 Participant flow diagram.

Instruments

Defeat scale

The Defeat Scale (D-Scale; Gilbert & Allan, 1998; Spanish adaptation by Ordoñez, Cuadrado & Rojas, 2021) assesses the psychological construct of defeat present in the motivational phase from the IMV model. This self-administered instrument comprises 16 items rated on a five-point Likert scale from 0 (never) to 4 (always). The final score is the sum of all the items (range: 0–64), where the higher the score, the higher is the level of defeat. The Spanish version of the D-Scale has shown excellent psychometric properties (Ordoñez, Cuadrado & Rojas, 2021). For the current study, the Cronbach’s alpha value was 0.79.

Entrapment scale

The Entrapment Scale (E-Scale; Gilbert & Allan, 1998; Spanish adaptation by Ordoñez, Cuadrado & Rojas, 2021) assesses the psychological construct of entrapment from the IMV Model. The instrument consists of 16 items that are scored on a range from 0 (not at all like me) to 4 (extremely like me). The higher the score, the higher is the level of entrapment. Items 1–10 are designed to measure internal entrapment and items 11–16 external entrapment. The E-Scale has shown excellent levels of internal consistency, with Cronbach’s alpha for internal entrapment being 0.93 and for external entrapment 0.89 (Carvalho et al., 2013). In this study, a Cronbach’s alpha value of 0.86 was obtained for both entrapment scales.

PROMIS®-Depression

Depressive symptomatology was assessed using the PROMIS® item bank (Cella et al., 2010), which is composed of eight questions rated from 0 (never) to 4 (almost always). The higher the score obtained, the more depressive symptoms the participant presents. The cut-off point for the presence of a possible depressive episode is set at 16 points. The reliability indices of the scale are considered excellent (Cronbach’s alpha 0.96). The reliability index found in the current work was 0.82. The authors have permission to use this instrument from the copyright holders.

Perseverative thinking questionnaire

The Perseverative Thinking Questionnaire (PTQ; Ehring et al., 2011) is an instrument based on the definition of ruminative thoughts present in various mood disorders. It comprises 15 items scored from 0 (never) to 4 (always) that quantify the degree to which a person is prone to show this type of thinking. The PTQ has adequate psychometric properties (Cronbach’s alpha of 0.95). In this work, the scale obtained a reliability index of 0.93.

Risk indicator questionnaire

The Risk Indicator Questionnaire (RIQ; Guibert, 2002) is an instrument used in the prevention of suicidal behavior. It consists of 13 dichotomous (yes/no) response questions that provide an objective assessment of suicidal behavior, including psychosocial factors, whereby items are summed to create a total index. This instrument was previously validated in Spanish-speaking University students. The RIQ presents a Cronbach’s alpha of 0.89. In the current study, the reliability index obtained was 0.71.

Procedure

The study protocol was approved by the Clinical Research Ethics of the Integrated Care Management of Albacete (Record N° 2021-134). First, the project was presented to the management team and teaching staff of the collaborating faculties to obtain the necessary authorizations. After approval, the members of the research team visited the different classrooms during in-person activities to inform the participants about the project. They explained the aims of the study, answered any questions, and requested signed informed consent from the participants. All participants were provided with a participant information sheet explaining the study and the ethical requirements. The data were collected in group format by four psychologists in a single session lasting approximately 60 min conducted during the students’ class time in their classroom. The questionnaires were administered on paper. All the participants were informed about the suicide prevention services available at their universities and were provided with detailed information on how to access these services. They were also given the opportunity to directly contact the research team (consisting entirely of psychologists) to discuss their suicidal thoughts and, if necessary, to be referred to the most appropriate health professionals.

Data analysis

Statistical analyses were conducted using IBM SPSS Statistics 28 (SPSS, Inc., Chicago, IL, USA). The criterion for statistical significance was set at p ≤ 0.05. Logical and range tests were performed, along with data consistency checks. Following data review and cleaning, exploratory and additional analyses were carried out to categorize and transform the variables. The specific data analysis plan included, firstly, a descriptive analysis of the main study variables. Finally, using AMOS 26.0 software, a path analysis was carried out to evaluate, on the one hand, the relationship between the variables of defeat, entrapment, and suicide risk and, on the other, to analyze the moderating role of rumination in the relationship between defeat and entrapment.

In the path analysis, the maximum likelihood method was used to estimate the different parameters of the model. The following parameters were used to evaluate the model fit: Comparative Fit Index (CFI; Hu & Bentler, 1999) and Root Mean Square Error of Approximation (RMSEA; Browne & Cudeck, 1993). Following Hu & Bentler (1999), the model presents a good fit if the value of the CFI index is equal to, or greater than, 0.95 and the RMSEA is less than, or equal to, 0.08 and the χ2 test is not significant.

To evaluate the moderating effect of rumination on the relationship between entrapment and defeat, the variable of the interaction between rumination and defeat was calculated. This variable was then included in the model as a predictor of variables of internal and external entrapment. Subsequently, following the recommendations of Cohen et al. (2003), to interpret the moderating effects of the two predictor variables on the criterion variable of entrapment, a graph of the relationship between defeat (predictor) and entrapment (criterion) was established, with the levels of the moderator (rumination) being located one standard deviation below and one standard deviation above their mean. Additionally, we estimated the statistical significance of each of these two slopes (Aiken & West, 1991), which represent the simple effect of the predictor on the criterion at the two levels of the moderator variable (Aiken & West, 1991).

For the mediational analysis, a bootstrapping analysis was conducted, as it has been found to be one of the most robust methods to detect mediation between variables (Hayes, 2022). The bootstrapping process creates different samples from the initial sample, using a resampling technique. In this analysis, the aim was to assess whether the relationship between entrapment (external and internal) and suicide risk (direct effect) was mediated by defeat (indirect effect). To this end, using the bootstrapping procedure, 95% confidence intervals with bias correction were estimated in 2,000 different samples to calculate all direct and indirect effects. If the resulting confidence interval for each effect does not include zero, the effect is considered to be statistically significant.

Finally, to ensure that the variables in the model are differently related as regards sex group, a multi-group comparison was performed. To assess sex equivalence, it were performed comparisons of a model in which factor loadings and covariations may vary between sex groups (fully unconstrained) with models in which some relationships between variables are set to be equal between sex groups (partially constrained): (a) relation between defeat and external entrapment; (b) relation between defeat and internal entrapment; (c) relation between rumination and external entrapment; (d) relation between rumination and internal entrapment; (e) relation between defeat-rumination moderator and external entrapment; (f) relation between defeat-rumination moderator and internal entrapment; (g) relation between defeat and suicide risk; (h) relation between rumination and defeat; (i) relation between external entrapment and suicide risk; and (g) relation between internal entrapment and suicide risk. Additionally, the unconstrained model was also compared to a fully constrained model. The model comparisons were performed using a standard ‘decrement to χ2’ test, in which the respective goodness of fit (and degrees of freedom) of the two compared models was differentiated.

Results

Descriptive analysis

Table 1 shows the descriptive statistics for the main study variables. The results indicate there were statistically significant differences between sex groups for rumination and suicide risk variables: namely, with women scoring higher on both variable relative to men.

Table 1 Means and standard deviations for the main variables in the study by gender group.

Variable	Male group (n = 150)	Female group (n = 374)	Cohen’s d	
Defeat	11.83 (7.36)	12.97 (6.94)	0.16	
External entrapment	6.02 (6.10)	6.00 (5.89)	<0.01	
Internal entrapment	3.19 (4.22)	2.72 (3.72)	0.12	
Rumination	18.28 (9.44)	20.70 (10.82)	0.24*	
Suicide risk	2.53 (1.92)	3.29 (2.82)	0.32**	
Notes:

** p < 0.01. Statistical significance was set at p < 0.01, bilateral.

* p < 0.05. Statistical significance was set at p < 0.05, bilateral.

Path analysis

Separately for each sex group, a path analysis model with direct associations between both types of entrapment and suicide risk, defeat and internal and external entrapment, defeat and suicide risk, and rumination and the internal and external entrapment variables was evaluated. A direct association between the rumination-defeat interaction variable and both types of entrapment was also included to assess the moderating effect of rumination on the relationship between defeat and entrapment. Finally, the correlations between the defeat and rumination variables, the rumination-defeat interaction variable with rumination and defeat, and both types of entrapment were included in the model. The mediating role of defeat in the relationship between entrapment (internal and external) and suicide risk was also assessed (see Fig. 2).

Figure 2 (A) Evaluation of the role of defeat as a mediator in the relationship between entrapment (internal and external) and suicide risk in male group. (B) Evaluation of the role of defeat as a mediator in the relationship between entrapment (internal and external) and suicide risk in female group.

The fit indices for the model indicated a good fit in both sex groups (χ2(2) = 0.82, p = 0.665; CFI = 1.00; TLI = 1.02; RMSEA = 0.00, and χ2(2) = 3.20, p = 0.202; CFI = 0.99; TLI = 0.99; RMSEA = 0.04, for male and female participants, respectively). The estimated parameters (presented as standardized) are shown in Fig. 2 for each sex group. For the male participants, the results showed that the defeat variable was significantly associated to external entrapment. The results also showed that external entrapment, but not internal, was statistically related to the risk of suicidal behavior. Additionally, the defeat variable showed a statistically significant indirect effect on the risk of suicidal behavior through the variables of entrapment (standardized indirect effect = 0.19, 95% CI [0.08–0.32], p = 0.002). In other words, the increase in the defeat variable was indirectly related to a higher score on the indicator of risk of suicidal behavior through higher levels of entrapment. Regarding the female group, the results indicated that the defeat variable was significantly related to both internal and external entrapment and that both entrapment variables were statistically associated to the risk of suicidal behavior. Additionally, the defeat variable showed a statistically significant indirect effect on the risk of suicidal behavior through entrapment (standardized indirect effect = 0.15, 95% CI [0.09–0.22], p = 0.004). Finally, in both sex groups, higher rumination scores were related to higher internal and external entrapment and had significant indirect effects on the risk of suicidal behavior through increased entrapment scores (standardized indirect effect = 0.34, 95% CI [0.20–0.48], p < 0.001, and standardized indirect effect = 0.17, 95% CI [0.12–0.23], p < 0.001 for males and females, respectively).

Finally, regarding the analysis of the moderating effect of the rumination variable on the relationship between defeat and both types of entrapment, the results showed that this type of thinking indeed exerts a moderating effect on the relationship between defeat and both types of entrapment in the female group, but not in the male group. Figures 3 and 4 shows the interaction of defeat and rumination in its relationship with the internal and external entrapment variables, respectively, in female participants.

Figure 3 Effect of the interaction between rumination and defeat on internal entrapment in the female group.

Figure 4 Effect of the interaction between rumination and defeat on external in the group.

The corresponding simple effects were calculated using the PROCESS macro for SPSS by Hayes (2022). The results showed that, for female participants with high (mean + 1 SD) levels of rumination, the association between defeat and internal entrapment was statistically significant (β = 2.93, p < 0.001, respectively), while this association did not reach statistical significance in females with low levels of rumination (β = 0.68, p = 0.064). That is, high levels of rumination strengthened the positive relationship between defeat and internal entrapment in females, but this was not true in the case of low levels of rumination. For external entrapment, the results showed that the association between defeat and external entrapment was statistically significant for both participants with low levels of rumination (mean − 1 SD: β = 1.82, p = 0.004), and those with high levels of rumination (mean + 1 SD: β = 3.70, p < 0.001). That is, the presence of rumination, regardless of the level, strengthened the positive relationship between defeat and external entrapment in females.

Regarding the multigroup comparison of the model, the results of the model testing and model comparisons are presented in Table 2. As can be seen, the unconstrained model showed good fit statistics, suggesting that a common path model could be assumed across sex groups. After that, it was assessed whether adding more restrictions to the model would improve it by making it more parsimonious and with better fit statistics. For this purpose, chi-square difference tests were performed comparing the unconstrained model with several partially constrained models. For each comparison, one at a time, only one specific path was constrained to be invariant across sex groups. The fully constrained and fully unconstrained models were also compared and the change in chi-square as a result of this comparison was significant.

Table 2 Comparative fit of tested models.

	Model fit indices		Model comparison tests	
Model	df	X 2	p	RMSEA	CFI	AIC	Δdf	ΔX 2	p (d)	
1. Fully unconstrained	4	4.02	0.403	0.00	1.00	104.02				
2. Partial unconstrained	10	11.53	0.117	0.02	1.00	99.57				
Model 1 vs. Model 2							7	11.53	0.117	
3. Fully constrained	13	30.25	0.004	0.05	0.99	112.25				
Model 1 vs. Model 3							9	26.23	0.002	

Finally, another model was developed in which all paths were constrained to be invariant across sex groups except for those specific paths that had been found to be statistically different in previous model comparisons: (1) the rumination to internal entrapment path (ΔX2(1) = 11.01, p = 0.001); (2) the moderator variable (RNT × defeat) to internal entrapment path (ΔX2(1) = 3.73, p = 0.050); and (3) the rumination to external entrapment path (ΔX2(1) = 6.87, p = 0.009). This partially constrained model also provided a good fit. The change in chi-square between the unconstrained and partially constrained models was not significant (see Table 2), indicating no detectable differences in the model, in which all but three paths were constrained to equality across age groups. Given the similarity between the statistics of both models, the Akaike Information Criterion (AIC) was used to decide which of the two models showed the best fit. Since the partially constrained model had a lower AIC value than the fully unconstrained model, the partially constrained model was selected as the model with the better fit.

According to the results obtained in the multi-group comparison, male and female groups showed statistically significant differences in the relationship between rumination and internal entrapment. This relationship was stronger in the male group (rumination explained 34.81% of the variance in internal entrapment), compared to the female group (19.36% of variance explained). Regarding the moderator variable and the internal entrapment, this moderation effect was higher in the female group (6.25% of variance explained) than in the male group (0.01% of variance explained). Finally, regarding rumination and the external entrapment path, in the male group, higher scores in RNT increased the external entrapment scores (29.16% of variance explained) while, in the female group, this association, although statistically significant, was lower than for males (15.21% of variance explained).

Discussion

University students are a risk group for suicide, with behaviours ranging from ideas and thoughts about suicide-to-suicide attempts of varying degrees of severity, and fatal suicide (Han et al., 2018; Hong et al., 2022; McKay et al., 2023). Research has shown that suicidal ideation and suicide attempts are more common among women, although more men than women die by suicide, commonly referred to as the sex paradox (National Institute of Statistics of Spain (INE), 2021; Shelef, 2021). In relation to sex approaches, descriptive results of this study showed that women reported significantly higher levels of rumination and suicide risk compared to men. Indeed, rumination has been suggested as a motivational moderator of the IMV model and a suicide risk factor (Rogers & Joiner, 2017). Furthermore, previous findings have also reported that this tendency to ruminate is higher in women than in men (Espinosa, Martin-Romero & Sanchez-Lopez, 2022; Johnson & Whisman, 2013). On the other hand, higher levels of rumination in females have been associated with higher levels of depressive symptomatology (Gomez-Baya et al., 2017), with this being posited as a possible explanation for the risk associated with sex differences, which emerge during adolescence and persist into adulthood (Gomez-Baya et al., 2017; Shors et al., 2017).

Regarding the dimensions of defeat and entrapment in the IMV model, no statistically significant differences were found between men and women in our sample of university students. This finding is consistent with previous studies reporting no sex differences for these dimensions (e.g., Lucht et al., 2020). Nonetheless, it is worth noting that while there is considerable knowledge on the impact of sociodemographic variables on suicidal thoughts and behaviors, the works analyzing sex differences in defeat and entrapment are scant and inconclusive. For example, Cramer, Rasmussen & Tucker (2019) reported higher levels of internal entrapment for females compared to males, but only for a sub-sample from United Kingdom, with such differences being absent in their U.S. participants. In a similar vein, the study by Ren et al. (2019) also described higher levels of entrapment in females than in males, although studying these differences was not an objective of the work. Given the robust link between the dimensions of the IMV model and suicidal thoughts and behaviors, it would be reasonable to assume similar sociodemographic patterns, although this hypothesis requires a deeper analysis to understand how these dimensions may differ across demographic groups (O’Connor & Portzky, 2018).

The first aim of the current study was to analyze the possible direct and indirect effects (through entrapment variables) of defeat on suicide risk. In accordance with the premises of the IMV model, the results obtained supported the notion that the perception of defeat, together with the perception of being trapped, with no possible escape or potential of rescue, is a factor in thinking of ending one’s own life as a way to escape from such a situation (O’Connor, 2011). Thus far, the IMV model has not distinguished between internal and external entrapment. However, the results of the present study, together with previous empirical findings (e.g., Forkmann et al., 2018; Höller et al., 2022; Rasmussen et al., 2023), highlight the importance of distinguishing between these two components. Furthermore, the sex differences found for internal and external entrapment could be explained by the nature of these dimensions. In fact, the higher rates of suicide deaths among adolescent and young adult males may be associated with a higher prevalence of externalizing disorders and psychosocial risk factors (e.g., parental divorce, access to means). In contrast, although both internal and external entrapment predicted the risk of suicidal behavior in the female participants in our study, previous works have shown that women are more prone to internalizing aspects of suicidal behavior (Miranda-Mendizabal et al., 2019). Consequently, these results may be considered an initial approach to this field of study, contributing to the existing literature on the pivotal role of defeat and entrapment in the development of suicidal ideation and behavior in general, and the differentiation between internal and external entrapment.

Additionally, the results of the current study showed that the moderating role of rumination in the relationship between defeat and entrapment, proposed in the theoretical model put forward by O’Connor & Kirtley (2018), is only applicable in women. Thus, specifically, women with high levels of rumination show a stronger association between internal entrapment and defeat. In the case of external entrapment, this defeat-entrapment relationship is associated with the presence of rumination, regardless of the level of defeat felt by the woman. To the best of our knowledge, no previous studies have reported similar results. Hence, although rumination tends to be more common in women, the capacity of this thinking style to directly predict both levels of entrapment (internal and external) is greater in men than in women, as evidenced by the results of our multigroup analysis. This finding appears to suggest that, although men tend to be less likely to ruminate, when they do, this rumination has a greater capacity to directly increase levels of entrapment. In contrast, in the female group, the levels of entrapment are not only influenced by the direct effect of rumination, but also by its moderating effect on the relationship between defeat and entrapment, enhancing the strength of the association between these variables.

In summary, these findings suggest that the role of rumination in generating feelings of entrapment is more complex and intricate in the case of women than in men. This aspect should be taken into account when designing interventions aimed at mitigating the variables of defeat and entrapment and, consequently, suicide risk.

This work has several limitations. Firstly, the cross-sectional study nature prevents from making cause-effect inferences. Nevertheless, longitudinal data would yield more complete information on within-participant development in each of the variables analyzed as well as how interact over time. For this reason, as recommended by Stenzel et al. (2020), given the short-term variability of defeat and entrapment, future studies might include repeated measurements of these variables in real time to adequately capture the empirical relationships between them. Secondly, the data were collected individually in groups. This could induce biased responses due to social pressure, for example, underestimating the actual prevalence of suicidal ideation. This is especially important when the participants are young, given they are more susceptible to social desirability. For future studies, it is recommended to prioritize methods that maximize confidentiality, such as anonymous online surveys. Thirdly, regarding the external validity of the results, the data were collected from a convenience sample comprising only university students. Future studies are recommended to replicate this research in a sample that is more representative of the general adolescent and young adult population, and to homogenize study samples according to sex, since the sample size of our male group has been small compared to the female group. In addition, although this study focused on adolescents and young adults, it should be noted that the university sample included participants aged over 25 years (n = 18). This age discrepancy should be considered when interpreting the results, and future attention should be paid to this age variability within the university population. Additionally, as regards the instruments used in the study, first, it is worth noting that the use of the Risk Indicator Questionnaire limits our comparison with previous research on the IMV model. Future research should include a standardized instrument (Campos et al., 2023; Runeson et al., 2017), such as the Columbia-Suicide Severity Rating Scale (C-SSRS; Posner et al., 2011), and differentiate between suicidal thoughts and behavior. And, secondly, the reliability obtained for the questionnaires used to assess defeat and suicide risk is only of an acceptable level, and we thus cannot rule out the presence of biases in our results. Finally, this study focused on differences based on biological sex (male - female) and not on gender differences. The recent literature (e.g., Cramer et al., 2023), however, suggests there are differences in the variables of entrapment, defeat and suicidal ideation according to gender identity, in the sense that gender minority individuals (i.e., transgender and gender diverse) report greater internal and external entrapment, defeat, and suicidal ideation compared to cisgender individuals. Accordingly, future work should include gender minority groups to obtain a more accurate picture of the relationship between these variables and to enhance the generalizability of the results to different types of populations.

Conclusions

To sum up, the results showed significantly higher levels of rumination and suicide risk in the female group. Moreover, defeat and entrapment were significantly associated with the suicidal risk. In the male group, the increase in the defeat variable was indirectly related to a higher score on the risk of suicidal behavior through higher levels of external entrapment, while in the female group the defeat variable was a significant predictor of both internal and external entrapment, and both entrapment variables were significant predictors of the risk of suicidal behavior. Additionally, the defeat variable showed a statistically significant indirect effect on the risk of suicidal behavior through entrapment. In both sex groups, higher rumination scores were related to increased internal and external entrapment, although this association was stronger in the case of the men, and there were significant indirect effects on the risk of suicidal behavior through higher entrapment scores. Finally, the mediating role of rumination in the defeat-entrapment relationship was only present in the female group, suggesting that the role of rumination in variables associated with suicide risk is more complex in women than in men.

Taking into account the significance of suicide and the alarming data concerning the university population, our findings could be of use in the implementation of interventions for the early detection of risk cases and the reinforcement of protective factors, especially among women who report higher levels of rumination and suicidal risk. Interventions could include approaches that address rumination as a key factor in the relationship between defeat and entrapment, particularly in women. In this line, several authors suggest that rumination-focused therapy interventions could reduce suicidal ideation/behavior (e.g., Hensel, Forkmann & Teismann, 2024). In fact, interventions such as metacognitive therapy and rumination-focused CBT (Watkins & Roberts, 2020) have been successful in the treatment of depression symptoms. Further, a sex perspective on the IMV model may provide useful, additional information about suicide risk and its predictors. Specifically, rumination seems to play a different role in men and women, suggesting that interventions should be tailored according to sex to be more effective. Future investigations could examine how other emotional regulation strategies, both adaptive and maladaptive, are associated with suicide. In this regard, a recent systematic review and meta-analysis (Rogier et al., 2024) suggests a complex pattern of association between these strategies and suicidality, which could offer a more nuanced perspective. Additionally, the integration of approaches that analyze how different motivational factors interact and affect suicidal risk, taking into account sex differences and other relevant moderators, could also be considered (Souza et al., 2024). Such research could provide additional information that would allow for the development of more specific and effective interventions tailored to the specific needs of university students, in particular, and various population groups, in general.

Supplemental Information

Supplemental Information 1 Dataset.

Additional Information and Declarations

Competing Interests

Author Contributions

Human Ethics

Data Availability

The authors declare that they have no competing interests.

Inmaculada Nayara Silvestre Vidal conceived and designed the experiments, performed the experiments, analyzed the data, prepared figures and/or tables, and approved the final draft.

Marta Nieto conceived and designed the experiments, analyzed the data, prepared figures and/or tables, and approved the final draft.

Jorge Ricarte performed the experiments, authored or reviewed drafts of the article, and approved the final draft.

María Dolores Vizcaíno Alcantud performed the experiments, authored or reviewed drafts of the article, and approved the final draft.

David Hallford analyzed the data, authored or reviewed drafts of the article, and approved the final draft.

Laura Ros conceived and designed the experiments, analyzed the data, prepared figures and/or tables, and approved the final draft.

The following information was supplied relating to ethical approvals (i.e., approving body and any reference numbers):

The study protocol was approved by the Clinical Research Ethics of the Integrated Care Management of Albacete (Record N° 2021-134).

The following information was supplied regarding data availability:

The dataset is available in the Supplemental File.

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
