# Peer review of "Effects of defeat and entrapment on suicide risk in university population: the role of rumination and sex"

_PeerJ, doi:10.7717/peerj.18673_

## Round 0.1 · original submission · Major Revisions

After reading the manuscript, I think that this contribution would be beneficial for the field, but it still need work. Specifically, I need the Authors to address all the issues raised by the Reviewers, specifically R1. Moreover, I do agree with R4 on the use of the correct terminology given the lack of an experimental design to infer the causality between the variables.

Reviewer 1 ·

Basic reporting

Thank you for this article. It examines an important population that is at risk of suicide and uses a clear theoretical background.
The introduction starts with explaining that suicide prevention is an important public health priority. It is mentioned that university students are a particular risk group. A rate of 4% is mentioned. Is this concerning deaths by suicide or suicide attempts (line 37)? A rate of 4% seems extraordinarily high. Is it 4% or 4 out of 100.000? Please specify this number as it seems unlikely.
For the next sentence (line 37-40) please specify as well. Internationally it has been shown that while men die by suicide more than women, women attempt suicide more often. Is this different in Spain, or is the reporting not clear? Please elaborate.
Line 46-48: the greatest increase since when? And do you mean internationally? For example in Belgium the greatest increase in suicide deaths in 2022 has been in women from 60 to 74 years old. Please be more detailed and nuanced in reporting these type of statements.
Line 82: I think there is a typo in this sentence (‘need’ is too much)
Line 97-99: it is stated that no previous study have examined this. Upon a quick search I found a review that looked at studies examining the IMV model (https://pubmed.ncbi.nlm.nih.gov/38626312/). In this review, multiple studies that examined the relationship between entrapment and rumination and other aspects of the IMV model (eg Li et al 2020, Dhingra et al 2016a; Teismann & Forkmann 2017).

Experimental design

The research question in the abstract is phrased as ‘anayzing the effects of defeat and entrapment on suicide risk’ (line 20-21). As this concerns cross-sectional data at one moment in time, no causal conclusions can be made. Instead the research question should be adapted to ‘analyzing the association between defeat and entrapment, on one side, and suicide risk, on the other side’. Similarly, the results in the abstract imply causation while this can not be concluded from the current study (‘defeat and entrapment were significant predictors of suicide risk’; line 25).
Same remarks for the first aim stated at the end of the introduction (line 106-107).
Please provide a detailed participant flow (line 120-122), with standardized flow diagram such as the CONSORT diagram.
Line 168-169: The assessments of participants was conducted in a group format. This seems to be a very important limitation, especially among younger individuals who are more susceptible to social desirability bias. Please elaborate on this important limitation in the discussion. Why was this decision made?
Line 205-209: Please elaborate more specifically on what ‘multi-group comparisons’ entailed and which statistical tests were used for this.

Validity of the findings

The discussion provides a clear summary of the results and links this to previous research in an appropriate manner. The conclusion should be nuance on some point to avoid implying causality (eg ‘defeat and entrapment significantly predicted suicidal risk’; line 374).

Additional comments

A general comment about the introduction: be cautious to be nuanced enough and report rates, numbers and trends in a correct manner. Additionally, be careful to distinguish suicide attempts, death by suicide and suicidal thoughts, as these are not the same and prevalences might differ for each of these.
The paper examines the possible moderating effect of gender. Gender seems to be defined binary as man or woman. Were non-binary individuals excluded from the study/analyses or was it based on sex rather than gender? Considering the younger population, where non-binary genderidentities are more and more present, it seems faulty to not include these in the study. I can understand that this group can be excluded for certain analyses as the lower number of non-binary individuals could result in a lack of statistical power for this group, but this decision should at least be mentioned and explained in the method and elaborated on in the discussion as important limitation.
The informed consent form (ICF) is provided, but is very limited. Almost no information is given about the aims of the study, rights and obligations of the participants, confidentiality, data processing,… Moreover, the inclusion and exclusion criteria are not properly stated on the ICF. The study was approved by the ethical committee so I assume it is not a problem, but it might be beneficial to consider a more extensive informed consent in the future.

Reviewer 2 ·

Basic reporting

Dear authors,
thank you for this insightful manuscript on defeat, entrapment and suicide risk. Your perspective on rumination and gender is very interesting, and the article illustrates a complex theory. Nevertheless, it would be helpful for the reader to gain a more in-depth theoretical background. Some questions remain about the methodology and further limitations should be added. I believe that the study on a non-clinical sample contributes to the literature but requires some minor revisions.

1) Language: The article is clear, and the English is very good. However, the theoretical background could be smoothed out a bit (e.g., l.48ff, “adolescence is a time...”, l.79f “This... This...”). It is a matter of writing style choice to use “we” and “us” (e.g., l.162, l-173, l.363) instead of the passive-voice constructions. For the Method section, a third person or passive style would be recommended to remain objective.

2) Literature references: The theoretical background is summarized in detail in the Discussion section, but it would be helpful if the previous literature were more clearly described in the Introduction section as well. At the time of writing, the paper by Ramos-Vera et al. (2024) on sex differences in entrapment in a multinational sample had not yet been published. Still, it would contribute to the current state of research if you cited this paper in your revision. (https://doi.org/10.3389/fpsyt.2024.1321207)

Further, some previous studies have not been mentioned, that could be of relevance, e.g.:
- Tucker et al. (2016): https://doi.org/10.1080/13811118.2016.1158682
- Teismann & Forkmann (2015): https://doi.org/10.1002/cpp.1999
- Cramer et al. (2023): https://doi.org/10.1080/00223891.2023.2220400
- Rasmussen et al. (2023): https://doi.org/10.1111/sltb.12990
- Wang et al. (2022): https://doi.org/10.21203/rs.3.rs-1222696/v1
- Höller et al. (2022): https://doi.org/10.1371/journal.pone.0270985

3) Figures: The labelling of the Y-axis does not match the information in Figures 2 and 3. Please check whether you are referring to internal or external entrapment. The title of Figure 2 refers to external entrapment, while the Y-axis and the note refer to internal entrapment. The title of Figure 3 is the same, but the Y-axis refers to internal entrapment and the note refers to external entrapment. Please clarify.

4) Hypotheses: Your research question is formulated exploratively, and you state the aims of the article (l.106ff). However, there is existing literature on the first question (see above), so hypotheses could be formulated regarding the effects of defeat on suicide risk/ideation.

Experimental design

Strengths: The research question is complex and fills a knowledge gap in relation to gender and rumination. It is relevant for the high-risk population of university students. The methods section describes the study procedure in detail. The authors obtained ethical approval.

Weaknesses or general questions:
1) Instrument: To my knowledge, most studies examine the trajectory from defeat and entrapment to suicidal ideation, as described in the IMV model (O’Connor, 2011; O’Connor & Kirtley, 2018). Assessing suicide risk as an outcome variable is a different approach and is not directly consistent with previous research. Could you please explain why you decided to use the instrument CIR (Guibert, 2002) to assess suicide risk (l.155ff)? For replication purposes, it would be helpful to use a standardized instrument in the future (Campos et al., 2023; Runeson et al., 2017), such as the Columbia-Suicide Severity Rating Scale (C-SSRS; Posner et al., 2011), and differentiate between suicidal thoughts and behavior.

Additional References, that are not in the paper:
- Campos et a. (2023): https://doi.org/10.1037/neu0000850
- Runeson et al. (2017): https://doi.org/10.1371/journal.pone.0180292
- Posner et al. (2011): https://doi.org/10.1176/appi.ajp.2011.10111704

2) Exclusion criteria: Ramos-Vera et al. (2024) state that the relevance of entrapment “lies in its strong connection with depressive disorders and suicide-related behaviors”. Was your purpose to investigate a non-clinical sample? Please state why you have excluded the students with clinically relevant depressive symptoms (l.116f), especially with the reference in your Discussion (see l.309ff).

3) Ethics: How was the high risk of suicide managed: Were participants given the opportunity to talk to a professional about their suicidal thoughts?

Validity of the findings

Thank you for providing all data. The Discussion section is detailed and well-structured. I suggest adding the following points on limitations and outlook:

1) It makes sense to consider all age groups in a population of university students. Your sample includes n = 18 participants over the age of 25. Still, you refer to the high-risk population of adolescents (i.e. 10 to 19 years) and young adults (i.e. 18 to 25 years) in your Introduction and Discussion. This aspect should be considered in the limitations (l.368f).

2) Another limitation is that gender was measured in binary terms. In a recent study, Cramer et al. (2024) state that “gender minority (i.e., transgender and gender diverse) persons reported higher internal and external entrapment, defeat, and suicidal ideation compared to cisgender persons”. https://doi.org/10.1080/00223891.2023.2220400

3) As an outlook, it could be added that repeated measurement of defeat and entrapment in real-time should be carried out in future, as recommended by Stenzel, Höller et al. (2020). https://doi.org/10.3390/ijerph17134685

4) For clinical practice, it would be helpful to address suicide-specific and ruminative interventions (l.361), such as rumination-focused CBT (Watkins, 2018).

Additional comments

Some general comments:

1) The abstract summarizes to most important results. In the Method part, it would be clearer to add the total number of patients included in the final analysis, e.g., “a total of 524 students completed measures of...” instead of mentioning the full sample.

2) Could you mention the base rate of suicidality in the Descriptives?

3) L.520: Correct upper case “C” of O’Connor

Annotated reviews are not available for download in order to protect the identity of reviewers who chose to remain anonymous.

Reviewer 3 ·

Basic reporting

Article meets basic reporting standards. I should mention that English is not my native language, so I cannot say whether the English used throughout is completely accurate. But, language used was easy to understand and clear!

Sufficient context is provided in the introduction. The aims are clearly stated, but there are no hypotheses stated.

It is also unclear why most of the introduction is focused on adolescence, while the included study participants are university student 17 yrs and older. It would help if authors would explicitly state the age period that would define as "adolescence". Especially when pertaining to section in lines 41-53.

Experimental design

The study meets the aims and scope of the journal. It is also clear what the aims of the study are (as stated in 1. basic reporting) and how it fills a knowledge gap (addition of entrapment in relationship between defeat and rumination).

With regards to methods:
The Cronbach's alpha is only acceptable for some scales. It would be nice if this was reflected on the limitations of the discussion section. I feel like the questionnaires do require some additional reflection in general in the Discussion section of the paper. The questionnaire regarding suicidality seems to measure more broad aspects as well (for example the item: ¿Se siente molesto por no responder con igual intensidad a agresiones psicológicas y/o físicas que otras personas le hacen a usted?). This is more of a risk factor of suicidality than an aspect of suicidality per se. So, it would benefit the paper if this is reflected upon in the Discussion.

The methods are described with sufficient detail, but information to replicate would also include whether questionnaires were completed online or physically. Whether students had to travel to location or how was this organized?

The analyses are well described and accurate for the suggested aims.

Validity of the findings

Results are reported clearly and rigorously. All results are reported, regardless of impact, novelty or statistical significance.

Discussion and conclusion are linked to research questions and substantiated with literature. I feel like it would be good to consider that correlation does not equate causation. Authors state in lines 347-348 "Thus, specifically in this group, rumination bolsters the association between internal entrapment and defeat, but only at high levels of the latter.". However, it would be more accurate to state that women with high levels of rumination (....) rather than stating rumination bolsters. As we do not know what the effect of rumination is considering the cross-sectional nature of the study. This should also be looked at for the conclusion.

As stated previously (2. Experimental design) the limitations section requires more elaboration on several aspects.

Conclusions:
Lines 386-388 authors state "Taking into account the significance of suicide and the alarming data concerning the university population, our findings could be of assistance in the implementation of interventions for the early detection of risk cases and the reinforcement of protective factors."

This seems a bit of a vague statement and applicable to any research that looks at suicidality. Additionally lines 390-396 are also not very applicable to this research specifically.

Additional comments

N/A

Reviewer 4 ·

Basic reporting

Figure 2 and 3 look the same figure with a different y-axis title and they are somehow inconsistent with the text at p. 11.

The number of participants in the abstract (650) is not the same as in the Participants section (p. 7) where it is reported to be 524 (which is consistent with having 150 males + 374 females).

Experimental design

No comment

Validity of the findings

In the limitations the Authors acknowledge that "the cross-sectional study nature prevents us from
making cause-effect inferences", but they have labelled variables in theri models as "independent" and "dependent" throughout the whole paper. Perhaps they Authors might consider using "predictors" and "criterion", respectively.

Motivational moderators of the IMV model include rumination but also several others (see, e.g., Souza et al. 2024 https://doi.org/10.1080/17437199.2024.2336013) and a recent systematic review and meta-analysis suggested a complex pattern of assocation between adaptive and maladaptive emotion regulation strategies and suicidality(Rogier et al., 2024, https://doi.org/10.1037/bul0000415). The Authors might consider addressing this issues in their suggestions for future studies.

Additional comments

No comment

---

## Round 0.2 · accepted · Accept

I'm happy to inform you that both reviewers appreciated that you addressed all their concerns. R1 noticed a typo at line 117, please correct it.

Reviewer 2 ·

Basic reporting

The following changes have improved the basic reporting:
- correction of statistics
- further literature references to deepen the theoretical background
- correction of axis labelling in figure 3 and 4
- formulation of hypotheses

Experimental design

Reviewers' questions about the experimental design were adequately addressed and listed in the limitations.

Validity of the findings

The authors have changed the terminology appropriately so as not to falsely imply causality. Future implications were added on the basis of literature references.

Additional comments

Dear editor and authors,
After reviewing the rebuttal and revised manuscript, I have come to the conclusion that the authors have addressed the reviewers' comments thoroughly, and the revisions have significantly improved the manuscript. Accordingly, I recommend it for acceptance. I noticed only one typo in line 117: there is an extra space, and the author's name is misspelled - please check this.

Reviewer 4 ·

Basic reporting

The Authors adequately addressed my comments.

Experimental design

The Authors adequately addressed my comments.

Validity of the findings

The Authors adequately addressed my comments.

Additional comments

The Authors adequately addressed my comments.